METHODS AND RESOURCES

# BigBrain 3D atlas of cortical layers: Cortical and laminar thickness gradients diverge in sensory and motor cortices

Konrad Wagstyl[1,2,3]*, Stéphanie Larocque[4], Guillem Cucurull[4], Claude Lepage[1], Joseph Paul Cohen[4], Sebastian Bludau[5], Nicola Palomero-Gallagher[5,6], Lindsay B. Lewis[1], Thomas Funck[1], Hannah Spitzer[5], Timo Dickscheid[5], Paul C. Fletcher[2], Adriana Romero[4,7], Karl Zilles[5], Katrin Amunts[5,8], Yoshua Bengio[4], Alan C. Evans[1]

1 McGill Centre for Integrative Neuroscience, Montréal Neurological Institute, Montréal, Canada, 2 Department of Psychiatry, University of Cambridge, Cambridge, United Kingdom, 3 Wellcome Trust Centre for Neuroimaging, University College London, London, United Kingdom, 4 MILA, Université de Montréal, Montréal, Canada, 5 Institute of Neuroscience and Medicine (INM-1), Forschungszentrum Jülich GmbH, Jülich, Germany, 6 Department of Psychiatry, Psychotherapy and Psychosomatics, Medical Faculty, RWTH Aachen University, Aachen, Germany, 7 Department of Computer Science, McGill University, Montréal, Canada, 8 Cecile und Oskar Vogt Institute for Brain Research, Heinrich Heine University Duesseldorf, University Hospital Duesseldorf, Düsseldorf, Germany

* k.wagstyl@ucl.ac.uk

**Data Availability Statement:** All code used for the segmentation and analysis of cortical layers are being made available at https://github.com/kwagstyl. BigBrain volumes, layer surfaces, and

## Abstract

Histological atlases of the cerebral cortex, such as those made famous by Brodmann and von Economo, are invaluable for understanding human brain microstructure and its relationship with functional organization in the brain. However, these existing atlases are limited to small numbers of manually annotated samples from a single cerebral hemisphere, measured from 2D histological sections. We present the first whole-brain quantitative 3D laminar atlas of the human cerebral cortex. It was derived from a 3D histological atlas of the human brain at 20-micrometer isotropic resolution (BigBrain), using a convolutional neural network to segment, automatically, the cortical layers in both hemispheres. Our approach overcomes many of the historical challenges with measurement of histological thickness in 2D, and the resultant laminar atlas provides an unprecedented level of precision and detail. We utilized this BigBrain cortical atlas to test whether previously reported thickness gradients, as measured by MRI in sensory and motor processing cortices, were present in a histological atlas of cortical thickness and which cortical layers were contributing to these gradients. Cortical thickness increased across sensory processing hierarchies, primarily driven by layers III, V, and VI. In contrast, motor-frontal cortices showed the opposite pattern, with decreases in total and pyramidal layer thickness from motor to frontal association cortices. These findings illustrate how this laminar atlas will provide a link between single-neuron morphology, mesoscale cortical layering, macroscopic cortical thickness, and, ultimately, functional neuroanatomy.

intensity profiles are available to download at http://bigbrain.loris.ca/ as well as at https://bigbrain.humanbrainproject.org. Tables with parcellated cortical and laminar thickness values are available in supporting information files S1 Data and S2 Data.

**Funding:** Parts of this work have received support from Healthy Brains for Healthy Lives, the Avrith MNI-Cambridge Neuroscience Collaboration award, the Wellcome Trust (215901/Z/19/Z), and the European Union's Horizon 2020 Framework Programme for Research and Innovation under Grant Agreement No. 785907 (HBP SGA2). The funders had no role in study design, data collection and analysis, decision to publish, or preparation of the manuscript.

**Competing interests:** The authors have declared that no competing interests exist.

**Abbreviations:** 5-HT2, 5-hydroxytryptamine/serotonin receptor 2; area OC, occipital area C; BZ, benzodiazepine; D1, dopamine receptor 1; GABA, gamma-aminobutyric acid; fMRI, functional MR; M2, muscarinic receptor 2; MEG, magnetoencephalography; NMDA, N-methyl-D-aspartate.

## Introduction

The cerebral cortex has laminar cytoarchitectonic structure that varies depending on cortical area [1] and cannot readily be resolved using in vivo MRI techniques [2]. Nevertheless, cortical microstructure underpins the functional, developmental, and pathological signals we can measure in vivo [3,4]. Thus, bridging the gap between microscale structural measurement and whole-brain neuroimaging approaches remains an important challenge. To address this, we sought to create the first whole-brain, 3D, quantitative atlas of cortical and laminar histological thickness.

Cortical thickness is one widely used marker of both in vivo and ex vivo cortical structure [5–7]. Early histological studies noted marked interareal thickness differences on postmortem histological sections [1,8], which have since been replicated [5,9] and extended using in vivo MRI [10], and alterations in these patterns may be seen in neuropsychiatric illness [11–13].

MRI approaches have demonstrated patterns of cortical thickness relating to functional and structural hierarchical organization across visual, somatosensory, and auditory cortices of both macaques and humans [10]. Although classical studies of cortical histology also observed that primary sensory regions are thinner than their surrounding secondary sensory cortices [1,8], the thickness gradients identified in MRI extend far beyond neighboring secondary areas into association cortical areas, whereas such a pattern has not been systematically studied in postmortem brains. However, MRI thickness is known to be impacted by the degree of cortical myelination [6,14], and cortical myelination exhibits similar gradients, with primary sensory areas being more heavily myelinated than secondary sensory areas [15]. Thus, it remains unclear whether thickness gradients found in MRI are artefactual, driven by gradient differences in cortical myelination causing systematic cortical reconstruction errors, or truly represent the underlying histology.

Creating a cortical layer segmentation of the BigBrain, a 3D histological model of the human brain [16], offers a solution to these problems and allows us to create a link between laminar patterns and standard MRI measures. Using this data set, we can determine whether cortical thickness gradients are evident in measurements made with much greater spatial resolution. It opens the possibility to study whether similar cortical thickness gradients are present in motor-frontal cortices such as those identified in in vivo neuroimaging [17]. Going beyond overall cortical thickness, it becomes possible to examine which cortical laminae contribute to these thickness gradients, enabling better characterization of cortical structure and the potential to link these macroscale thickness gradients to changes in laminar cortical connectivity in sensory and motor hierarchies.

Sensory processing hierarchies describe the concept that the cerebral cortex is organized with gradually changing structural and functional properties from primary sensory areas, to secondary sensory areas, and, ultimately, higher-order association areas. Multiple measurement modalities converge on similarly ordered patterns, including increasing dendritic arborization in of pyramidal neurons [18] and electrophysiological characteristics [19], laminar connectivity patterns of projecting cortical neurons [20–22], laminar differentiation [23,24], MRI cortical thickness [10], MRI myelination [15], receptor densities [25], and temporal dynamics [26]. Topographically, hierarchies are organized such that progressively higher cortical areas are located with increasing geodesic distance (the shortest path traversing the cortical surface) from their primary areas [10,27]. Ordering cortical areas along these gradients provides a framework for quantifying and understanding the relationships between cortical topology, microstructure, and functional specialization.

Carrying out analyses of histological thickness gradients poses several methodological challenges. First, thickness measurements carried out in 2D are associated with measurement

artefacts due to oblique slicing [28], and stereological sampling bias as out-plane cortical areas cannot be included. Second, manual measurement is associated with observer-dependent variability, estimated to be up to 0.5 mm [8]. Third, because of the labor-intensive nature of histological analysis, many histological atlases have a small number of sample points, with studies commonly restricted to measuring around 100 cortical samples [8,29]. These factors hinder the ability to detect and map potentially subtle cross-cortical variations in cytoarchitecture as well as overall and laminar thicknesses. BigBrain offers a unique data set to resolve histological cortical layers comprehensively in 3D, thereby providing a concrete link between microscale patterns of structure and in vivo markers.

We therefore set out to automate segmentation of cortical layers in 3D in order to characterize patterns of cortical and laminar thickness across visual, somatosensory, auditory, and motor-frontal cortical areas. To do this, we used a convolutional neural network to segment profiles of histological intensity sampled between the pial and white matter. Training profiles were generated from examples of manually segmented layers on cortical regions from 2D histological sections of the BigBrain data set. The trained network was used to segment intensity profiles derived obliquely through the 3D histological volume and generate mesh segmentations of 6 cortical layers. These surfaces were used to calculate cortical and laminar thicknesses. Geodesic surface distance from primary visual, auditory, somatosensory, and motor areas were calculated and used as a marker of hierarchical progression. Cortical and laminar thickness gradients were calculated for each system.

## Results

The automatically identified cortical layers closely follow bands of intensity within the Big-Brain (Fig 1) and continue to follow the same features beyond the limits of training examples (Fig 2A).

In the original BigBrain surfaces, as with MRI white matter surfaces, the white surface was placed at the maximum intensity gradient between gray matter and white matter [30]. By contrast, the neural network is trained on examples in which the white boundary has been manually located according to the presence of cortical neurons. This has caused a systematic shift in the location of the new white matter surface. On closer inspection, the maximum gradient at which the original surfaces were placed appears to be at the border between sublayers VIa and VIb, where the change in neuronal density is sharper than at the boundary between white matter and layer VI (S4 Fig).

A second feature apparent on visual inspection is segmentation of the layers cannot follow a single set of rules applied indiscriminately—laminar segmentations vary between cortical areas. This is most readily apparent at the V1-V2 boundary, where layer IV changes considerably (Fig 2B). Layer IV is particularly broad in V1 and has multiple sublayers creating extra peaks and troughs in the intensity profiles, whereas in V2, it is much thinner and no longer differentiated into sublayers. The transition from a thick layer IV to a thin layer IV occurs precisely at the boundary between these 2 regions, suggesting that the network is also internally learning certain areal properties.

### Comparison of total and layer thickness maps

On visual inspection, maps of BigBrain cortical thickness are consistent with classical atlases of histological thickness reported by von Economo and Koskinas (Fig 3B). In particular, the precentral gyrus is the thickest part of the cortex, with values over 4.5 mm (when adjusted for shrinkage in BigBrain) and 3.5 to 4.5 mm in von Economo (area FA). The thickness of the motor cortex is often underestimated in MRI thickness measurement [31], probably because

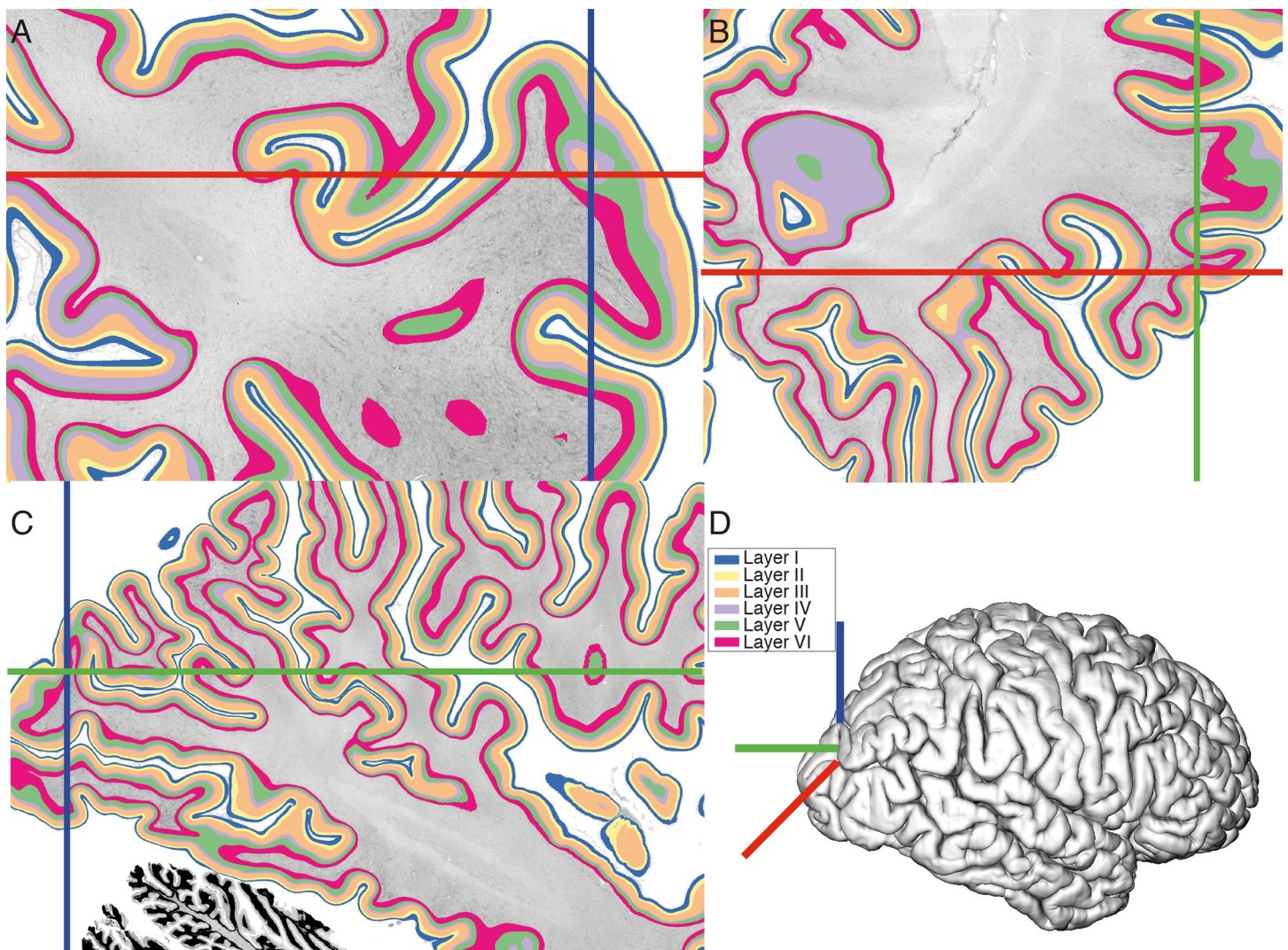

**Fig 1. Cortical layers in 3D.** Six cortical layers segmented on the 3D volume on 3 orthogonal planes: A = coronal, B = axial, C = sagittal. Panel D shows the location of the sections on the reconstructed pial surface of the 3D BigBrain. (A) The coronal plane is the original plane of sectioning. Within this plane, the axes are centered on an area of the cortex where layers would be impossible to segment in 2D because the section only shows part of the gyrus, and most layers are not visible because of the oblique sectioning of the cortex with respect to the gyrus. Underlying data available from ftp://bigbrain.loris.ca.

of the high degree of intracortical myelination that affects the gray–white contrast, causing the white matter surface to be too close to the gray surface, such that cortical thickness is underestimated [10,14]. The calcarine sulcus is especially thin on both BigBrain (1.67 to 2.86 mm, 95% range) and von Economo (1.8 to 2.3 mm, occipital area C [area OC]). This is also consistent with measurements from Amunts [32] of 1.47 ±. 24 mm (left) and 1.57 ± 0.41 (right). Overall, regional values from BigBrain were highly correlated with their corresponding values in von Economo and Koskinas (left hemisphere: r = 0.86, right hemisphere: r = 0.86). In addition, folding-related differences are clearly visible on the BigBrain, with sulci being thinner than their neighboring gyri. Vertices located in the medial wall and temporal lobe cuts were masked for all analyses of cortical thickness, and allocortex was additionally excluded for analyses of laminar thickness.

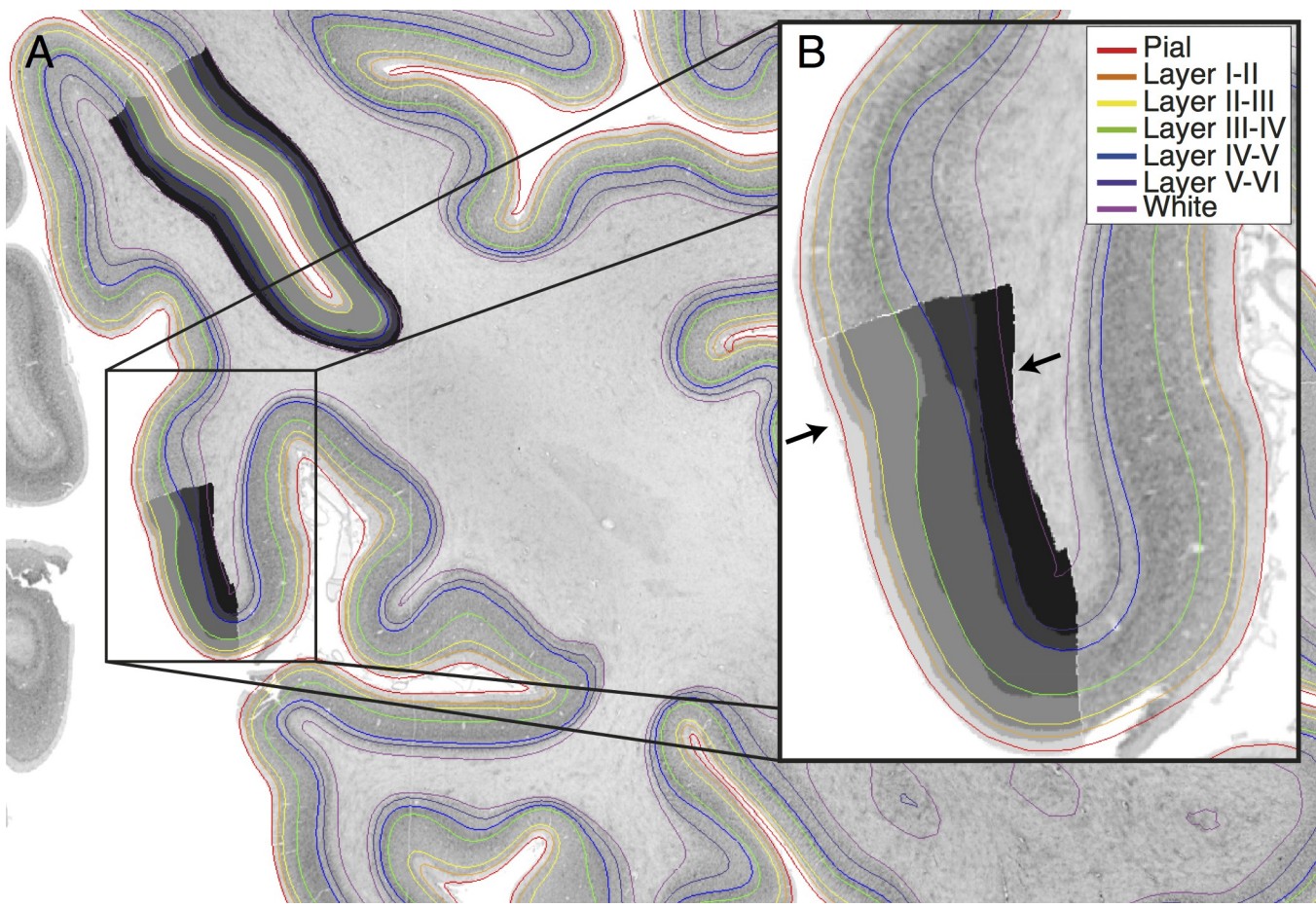

**Fig 2. Cortical layers (colored lines) intersected on a 2D coronal section of the right occipital cortex with manually segmented layers (superimposed grayscale masks).** (A) The boundaries follow the same contours as delineated by the manually segmented training areas and appear to accurately follow the layer bounds outside of each training area. (B) At the V1–V2 boundary (marked with arrows), the thickness of layer IV changes dramatically in both manual and automated segmentations (between green and blue lines), with additional peaks in V1 intensity due to the sublayers of layer IV. As each profile is individually segmented by the network, without reference to the neighboring profiles, the network is able to apply area-specific rules according to the shape of the profile, suggesting it might be internally identifying the area from which the profile is extracted as being either V1 or V2. Underlying data available from ftp://bigbrain.loris.ca.

BigBrain histological thickness values also closely correlated with regional values of MRI cortical thickness (Fig 3C). Overall regional values were highly correlated (left hemisphere: r = 0.62, right hemisphere: r = 0.75). Differences may relate to individual variability, age differences, and modality-specific biases. In a ranking of the differences between measurements, MRI cortical thickness was thinner than expected for heavily myelinated primary sensory areas (LBelt, 3b and V1), but thicker than expected for insular and peri-insular regions. This may be as the cortex is thin but heavily convoluted, with sulci that are fused difficult to resolve using MRI. Thickness used in the histological and MRI comparisons can be found in S1 Data, S2 Data.

BigBrain layer thickness maps are also consistent with layer thicknesses from the von Economo atlas (Fig 4). Layer III is thick in the precentral areas, and particularly thin in the primary visual cortex. Layers V and VI are thicker in frontal and cingulate cortices, but also thin in the occipital cortex. Each layer is strongly correlated with the corresponding von Economo measurements except layer II (layer I, left r = 0.50, right r = 0.46; layer II, left r = 0.11, right

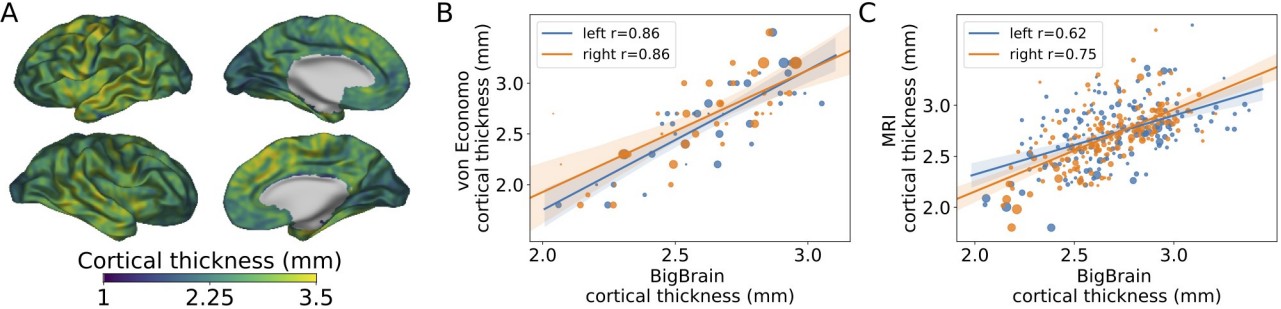

**Fig 3. Comparison of cortical thickness from the BigBrain with von Economo and Koskinas histological measurement and MRI cortical thickness data from the Human Connectome Project [33].** Thickness values range from 1.8 mm in the calcarine sulcus to 4.5 mm in the precentral gyrus. (A) Per-vertex cortical thickness values from the BigBrain (displayed on smoothed surfaces, values were smoothed 3 mm FWHM). Thicker regions of the cortex included the precentral gyrus containing the primary motor cortex. The occipital cortex around the calcarine sulcus was particularly thin. Also visible are smaller-scale variations in thickness than can only be observed through such high-density measurement. Von Economo reported thickness measurements from around 50 cortical areas, whereas the thickness of around 1 million vertices has been measured on BigBrain. (B) Regional BigBrain thickness values were highly correlated with measurements from von Economo and Koskinas. The size of each point is proportional to the area of the cortical region, and overall correlations were weighted according to these areas. The precentral gyrus, area FA, was the area of greatest discrepancy where BigBrain provided a lower estimate than von Economo. This might, in part, have been due to averaging of many vertices across the precentral gyrus in BigBrain, in comparison to a single measurement made by von Economo. (C) Regional BigBrain thickness values were also highly correlated with MRI cortical thickness values. MRI thickness appears to be overestimated in the insula, where it is thin in both histological data sets. This may be as the insula is highly convoluted and thus challenging to accurately delineate at lower resolutions. Underlying data available from S1 Data, S2 Data, and ftp://bigbrain.loris.ca. Area FA, frontal area A; FWHM, full width at half maximum.

r = 0.06; layer III, left r = 0.74, right r = 0.72; layer IV, left r = 0.80, right r = 0.76; layer V, left r = 0.66, right r = 0.66; layer VI, left r = 0.66, right r = 0.60). The reason for the lack of agreement between layer II measurements might arise from known difficulties in distinguishing a clear boundary between layer II and layer III alongside a low amount of interareal variation in layer II thickness. Similar challenges exist in identifying layer IV and the layer V–VI boundaries in many cortical areas, which may account for some of the discrepancies between the 2 sets of laminar measurements. These difficulties might be reflected in the lower "confidence" values associated with these areas (S2 Fig).

Nevertheless, despite the challenges associated with manual laminar segmentation and the fact that measurements were made on different individuals nearly a century apart, there are high overall correlations between these 2 laminar atlases.

Of additional interest is the clear boundary exhibited by layer IV at the boundary between V1 and V2 in the occipital cortex. This change in thickness is clear enough to generate an automated anatomical label for V1 (Figs 2 and 4).

## Cortical gradients and processing hierarchies

Cortical thickness was positively correlated with geodesic distance in visual (left, r = 0.57, p = 0, right, r = 0.44, p = 0; von Economo r = 0.72, p = 0.02), somatosensory (left, r = 0.21, p = 0, right, r = 0.28, p = 0; von Economo r = 0.64, p = 0.12), and auditory cortices (left, r = 0.18, p = 0, right, r = 0.12, p = 0; von Economo r = 0.92, p = 0.01) (Fig 5A–5C). By contrast in the motor cortex, thickness was negatively correlated with geodesic distance (left, r = −0.36, p = 0, right, r = −0.25, p = 0; von Economo r = −0.84, p = 0) (Fig 5D). These results are consistent with MRI thickness findings in sensory gradients but contradictory for the motor-frontal gradient.

Cortical layers did not contribute equally to the total thickness gradient in the visual and somatosensory cortices (Fig 6A). Layers III and V had the largest contributions to the total thickness gradient, followed by layer VI, and then II. A similar but less pronounced pattern was seen within the auditory cortex. In the motor cortex, the inverse was true, with decreases

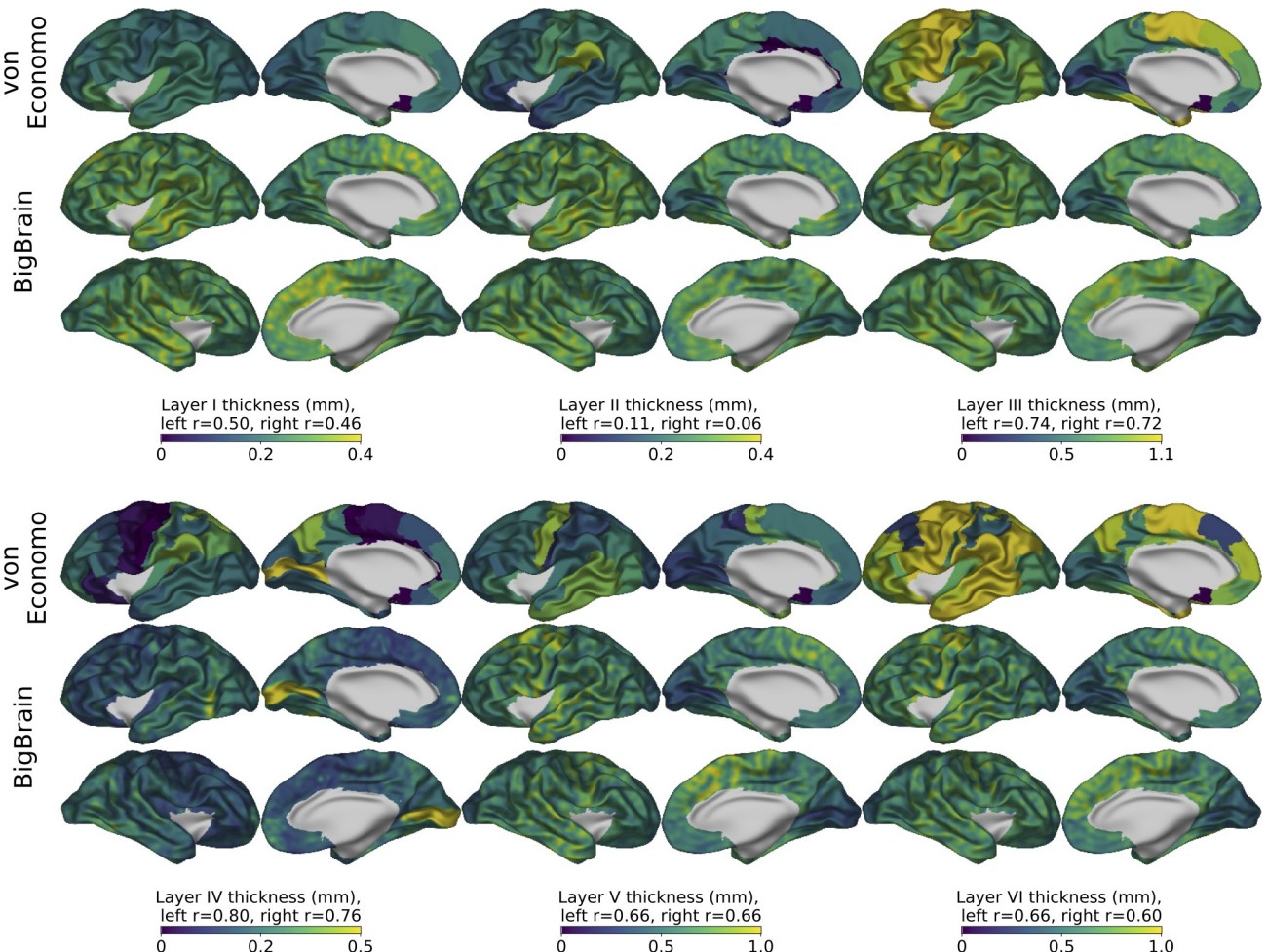

**Fig 4. Comparison of von Economo's laminar thickness maps (coregistered with and visualized on the BigBrain) with laminar thicknesses of the BigBrain for left and right hemispheres.** BigBrain thickness values were smoothed across the surface with a 3-mm FWHM Gaussian kernel. Layer thickness values strongly correlated between BigBrain and von Economo for all layers except layer II (see Results). Similarities include the clear changes in thickness in pre- and postcentral thicknesses of layers III, V, and VI. For layer IV, the most striking feature is the abrupt change in layer IV thickness at the V1–V2 border. This abrupt change and the unique features of layer IV in V1 lead us to conclude that the neural network may have internally learned to recognize V1 and apply the appropriate laminar segmentation rules. Underlying data available from S1 Data and ftp://bigbrain.loris.ca. FWHM, full width at half maximum.

in layers III, V, and VI. Changes in the same cortical layers appeared to drive gradients in the von Economo laminar thickness measurements (Fig 6B), but because of the small number of recorded samples, the confidence intervals were larger and generally included zero.

Thus, visual, auditory, and somatosensory areas exhibited positive histological thickness gradients primarily driven by layers III, V, and VI. By contrast, the motor-frontal areas exhibited an inverse gradient, peaking in the motor cortex and driven by the same layers (Fig 6B and 6C). Underlying data are available from S1 Data and ftp://bigbrain.loris.ca.

## Neural network training

In the cross-validation, average per-point accuracy on the test fold was 83% ± 2% prior to post-processing, indicating that the network was able to learn generalizable layer-specific features and transfer them to novel cortical areas. The predictions of the model trained on the full data

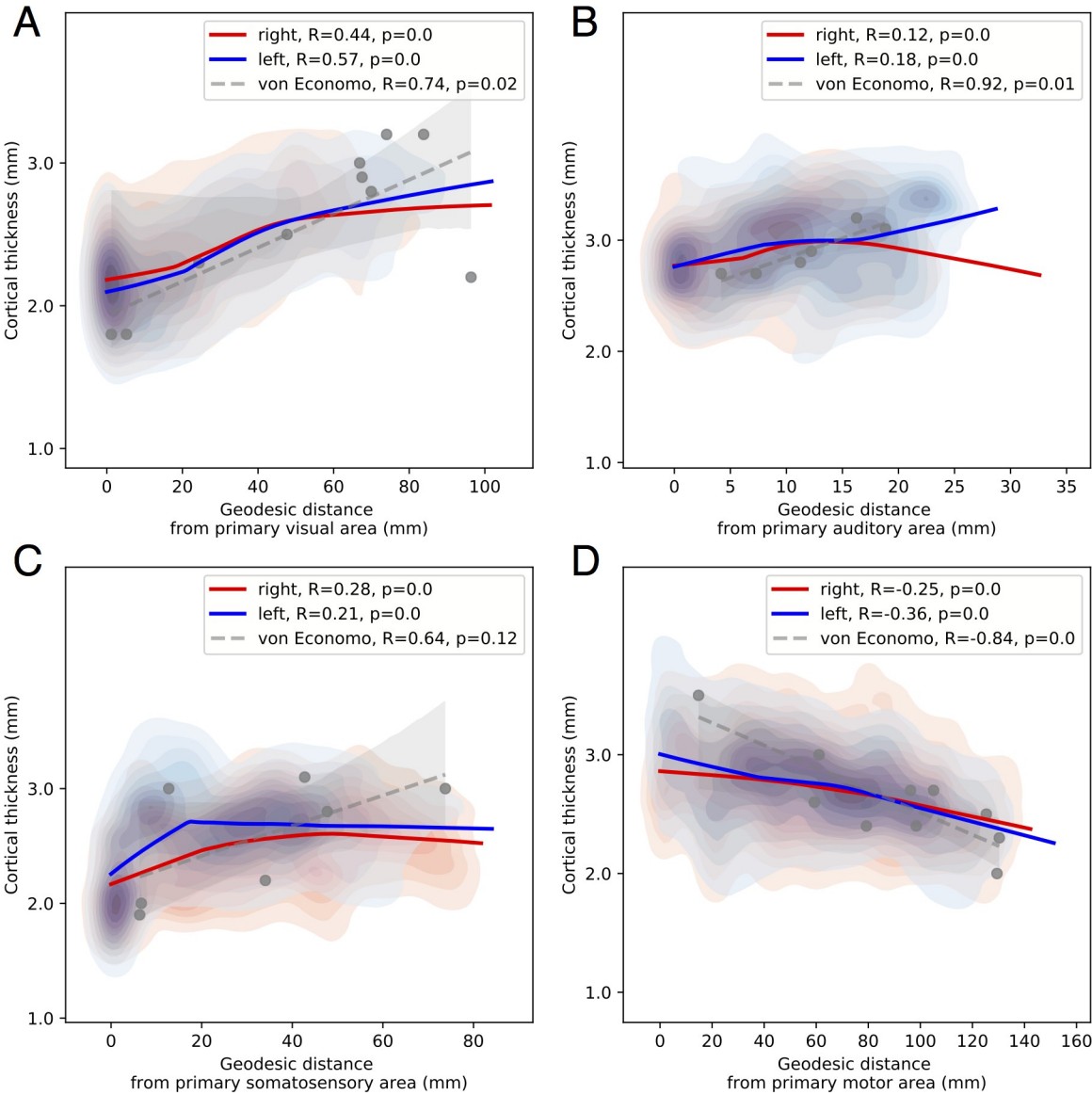

**Fig 5. Cortical thickness with increasing geodesic distance from the primary area.** To aid visualization, locally weighted scatterplot smoothing lines are fit for each hemisphere. For primary visual, auditory, and somatosensory cortices (A–C), consistent with MRI studies of cortical thickness, thickness increased with geodesic distance from the primary sensory areas. These trends were also present in the von Economo data set, where statistical power was limited by the small number of samples. For the motor cortex (D), a negative relationship was present with thickness decreasing from the primary motor cortex into the frontal cortex in the BigBrain data set and von Economo. This structural gradient is the inverse of the pattern of myelination and of previously reported MRI frontal thickness gradients but consistent with patterns of structural type and neuronal density. These findings suggest the presence of distinct but overlapping structural hierarchies. Underlying data available from S1 Data and ftp://bigbrain.loris.ca.

set were used to create a 3D segmentation of the cortical layers in both hemispheres of the Big-Brain data set (Fig 1).

## Confidence results

Layer confidence maps, given by the difference between prediction values (between 0 and 1) of the highest and second-highest predicted classes for each point, give an approximation of the

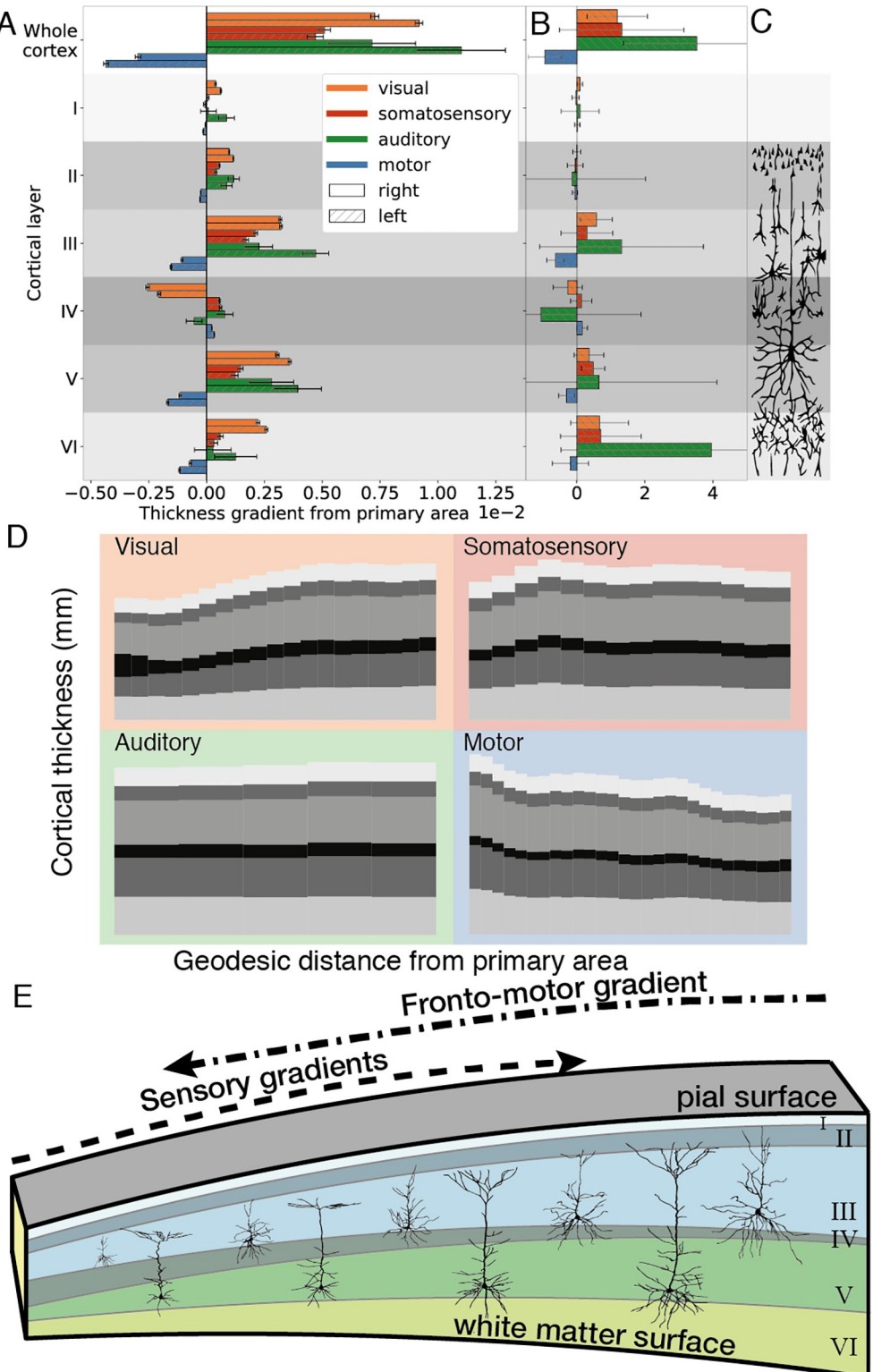

**Fig 6. Gradients of cortical and laminar thickness against geodesic distance from primary areas.** (A) Motor-frontal gradients show an inverse relationship from sensory gradients on both cortical and laminar thicknesses. Increasing sensory cortical thickness gradients were generally driven by thickness increases in layers III, V, and VI. By contrast, motor-frontal cortical thickness gradients exhibited decreases in thickness of the same layers. (B) The same trends were evident in the von Economo data set; however, because of the small number of recorded samples, the confidence

intervals were larger and generally included zero. (C) Typical neuronal types and morphologies of individual cortical layers. Cortical thickness gradients in either direction are primarily driven by changes in pyramidal cell layers (in layers III, V, and VI). (D) Layer thicknesses averaged across vertices in a sliding window of geodesic distance values from the primary area for the visual, somatosensory, auditory, and motor systems. The motor cortex exhibits the inverse pattern of change to those observed in sensory gradients. (E) Single-cell morphological studies of pyramidal neurons in macaque sensory processing pathways reveal increasing dendritic arborization [34] consistent with the hypothesis that laminar volume changes and ultimately thickness changes represent increases in intracortical connectivity. Underlying data available from S1 Data and ftp://bigbrain.loris.ca.

reliability of laminar segmentations for the cortex where ground truth manual segmentations have not been carried out (S2 Fig). Throughout the cortex, the network has high confidence for suprapial and white matter classes. Cortical layers also exhibit consistent confidence maps, with slightly lower confidence for layer IV. This pattern matches with visual observations that layer IV is often the most difficult layer to identify.

## Resolution results

Downsampling BigBrain to decrease the resolution, from 20 μm down to 1,000 μm, progressively decreased the accuracy of the network on the test folds from 85% to 60% (S3 Fig). However, at 100 μm (the approximate upper limit for current high-resolution structural MRI), profiles had sufficient detail to maintain an accuracy of 76%.

## Discussion

We automatically segmented the 6 histological layers of the cerebral cortex in the 3D reconstructed BigBrain. This is the first whole-brain quantitative, laminar atlas with high precision and the first ever in 3D. Our approach overcomes many historical problems with histological thickness measurements and provides a higher level of precision and detail than any past laminar atlases. We used this atlas to test for gradients of cortical and laminar thickness within sensory and motor processing hierarchies. Consistent with previous findings using in vivo MRI [10] and 2D histological measurements [8], visual, somatosensory, and auditory hierarchies exhibited a gradient of increasing cortical thickness from primary sensory to higher-order areas. These gradients were primarily driven by layers III, V, VI. By contrast, the motor-frontal cortices exhibited a decreasing cortical thickness gradient away from the primary motor cortex towards higher-order frontal areas, which was driven by decreases in these layers. These findings highlight the utility of the BigBrain for linking micro- and macroscale patterns of cortical organization.

Gradients of thickness are large-scale markers of systematic changes in the cortical histology. The volume of the cortex is 80% to 90% neuropil [35–39], of which 60% is axons and dendrites, and the remainder is synaptic boutons, spines, and glia. As neuronal density does not tend to correspond with increases in cortical thickness [40,41], and the majority of the cortical volume is made up of neuropil, increased thickness is most likely to indicate increased intracortical connectivity [7]. At a laminar level, the strongest contributors to the overall thickness gradients were layers III, V, and VI (Fig 6). Cell morphological studies in macaques have shown that the cell size and dendritic arborization of layer III and V pyramidal neurons increase along the visual pathway [18,42,43]. Similarly, afferent axonal patch sizes scale with pyramidal neuronal arborization [44]. Increases in dendritic arborization, axonal field size, and number of synapses would all give rise to an increase in the volume of laminar neuropil and are therefore plausible contributors in the laminar and overall thickness gradients measured here. Gradients of layer thickness provide us with a mesoscale link between in vivo patterns of MRI cortical thickness and histological changes in the cortical structure. Such links

help us to understand the neurobiological significance of interindividual, longitudinal, and neuropathological biomarkers [7].

In contrast to in vivo studies of motor-frontal functional, myelin, and MRI cortical thickness organization, which place the primary motor cortex at the same level as primary sensory areas [15,17,27], we found that total and laminar motor-frontal thickness gradients were the inverse of those measured in sensory cortices. This places the motor-frontal cortex in a distinct hierarchical position. Histologically, the motor cortex was especially thick, and the thickness decreased with geodesic distance from the primary motor cortex, with layers III, V, and VI following a similar inverse pattern. This finding is consistent with reported trends in other histological properties, such as laminar structural type [8] and neuronal density [40], as well as the observation that the motor cortex has large, pyramidal neurons with extensive dendritic arborization [45,46]. It is also in agreement with the distribution of neurotransmitter receptors. The molecular architecture as estimated by neurotransmitter receptors also provides evidence that primary visual and motor cortex are on opposite positions in cortical hierarchy—the acetylcholinergic muscarinic receptor 2 (M2), but also N-methyl-D-aspartate (NMDA), gamma-aminobutyric acid (GABAA), GABAA/benzodiazepine (BZ), $\alpha 2$, 5-hydroxytryptamine/serotonin (5-HT2), and dopamine (D1) receptors show high densities in the primary sensory areas, lower densities in association areas, and the primary motor cortex among the lowest [25].

Functionally, these structural differences might be considered in terms of narrow, specific columnar receptive fields for accurate sensory perception [47] and wider receptive fields [48] for the coordination of multiple muscle groups [49] in precise motor control. Such microstructural trends are likely to be a result of matching gradients of genetic expression [50] and may indirectly relate to other microstructural trends, including the relative somal size and connectivity patterns of pyramidal neurons in layers III and V [22,51]. Thus, there is a coherent group of cortical histological properties that diverges from patterns of cortical myelination and functional MRI (fMRI)-derived gradients, establishing the motor cortex at the peak of a gradient of increasing cortical thickness; layer III, V, and VI thickness; and pyramidal neuronal arborization, with primary sensory areas at the opposite extreme.

## Atlas of cortical layers

The layers we have generated to test gradient-based hypotheses have applications beyond the scope of this study. Surface-based models of layer structure also create a framework for translating between microstructural modalities and surface-based neuroimaging. For instance, layer segmentations can be used to define regions of interest for further detailed analysis and for associating cortical in vivo and ex vivo data to the common BigBrain template. Furthermore, current approaches to measuring laminar structure and function in vivo rely on prior models of the cortical layers—for example, signal-source simulation in magnetoencephalography (MEG) [52] or for laminar sampling in fMRI [53]. The whole-brain histological models for areal layer depth provided here, combined with a thorough understanding of how the layers vary with local cortical morphology [28,54,55], will aid such anatomical models.

## Limitations

It is important to acknowledge that the gradients of laminar thickness measured may be affected by limitations in the BigBrain data set. The first limitation is that the postmortem brain was damaged during extraction and mounting. In some areas, this resulted in minor shears. This problem was addressed to some extent through the utilization of nonlinear registration techniques. Nevertheless, some shifts in cortical tissue between consecutive sections are present and will affect the accuracy of layer reconstructions. In other areas, the cortex has been

torn. Spatial smoothing and the large total number of sample points make it unlikely that these errors are affecting these results. A second limitation is that there is only one BigBrain. Future work will be necessary to establish the interindividual and age-dependent variability in laminar structure, either using other histological BigBrains or with complimentary high-resolution MRI imaging approaches. Finally, at 20-μm resolution, individual neuronal cell bodies cannot readily be resolved. Future work to generate comprehensive histological atlases with even higher resolution will offer further insights into the direct links between mesoscale features presented here and microscale histological properties.

## Conclusions

Total cortical thickness and thicknesses for each of the 6 isocortical layers were measured in the BigBrain to explore the histological drivers of MRI-based thickness gradients. Overall, the pattern of thickness in the BigBrain is consistent with histological atlases of cortical thickness, such as that from von Economo and Koskinas [8]. In the visual, somatosensory and auditory cortices, an increasing gradient of histological cortical thickness was identified and found to be primarily driven by layers III, V, and VI. In the motor-frontal cortex, the inverse pattern was found. These findings provide a link between patterns of microstructural change and morphology measurable through MRI and emphasize the importance of testing MRI-based anatomical findings against histological techniques. The laminar atlases provide an invaluable tool for comparison of histological and macroscale patterns of cortical organization.

## Materials and methods

### Volumetric data preparation

BigBrain is a $20 \times 20 \times 20$ μm (henceforth described as 20-μm) resolution volumetric reconstruction of a histologically processed postmortem human brain (male, aged 65), in which sections were stained for cell bodies [56], imaged, and digitally reconstructed into 3D volume [16]. It is available for download at ftp://bigbrain.loris.ca and is used as a reference brain of the Atlases of the Human Brain Project at https://www.humanbrainproject.eu/en/explore-the-brain/atlases/. In order to run computations on this 1-TB data set, the BigBrain was partitioned into 125 individual blocks, corresponding to 5 subdivisions in the x, y, and z directions, with overlap. The overlap of blocks was calculated to be sufficient such that each single cortical column could be located in a single block, enabling extraction of complete intensity profiles between pairs of vertices at the edge of blocks without intensity values being altered by boundary effects when the data were smoothed. Blocks were smoothed anisotropically [57], predominantly along the direction tangential to the cortical surface, to maximize interlaminar intensity differences while minimizing the effects of intralaminar intensity variations caused by artefacts, blood vessels, and individual neuronal arrangement [28]. The degree of anisotropic smoothing is determined by repeatedly applying the diffusive smoothing algorithm, in which the degree of smoothing in a given direction is inversely related to the intensity gradient in that direction [57]. The optimal level of smoothing was previously determined and gave an effective maximum full width at half maximum (FWHM) of 0.163 mm [28]. For subsequent analyses, both the raw 20-μm and anisotropically smoothed blocks were used.

Lower-resolution volumes were extracted by subsampling the raw BigBrain 20-μm volume at 40, 100, 200, 400, and 1,000 μm. Anisotropically smoothed volumes were also generated at each of these resolutions.

## Profile extraction

Pial and white surfaces originally extracted using a tissue classification of 200 μm were taken as starting surfaces [30]. Each surface contained an equal number of surface points ("vertices"), located at corresponding anatomical locations on the 2 surfaces. Prior to intensity profile extraction, the vertex locations on both surfaces were altered to address several issues. First, the vectors connecting white and gray vertices were altered in order to improve their approximation of columnar trajectories and to minimize intersecting streamlines. Second, "pial" and "white" surfaces were, respectively, expanded beyond the pial boundary and into the white matter respectively, extending the extracted profiles to contain the whole cortex with additional padding. This was to enable the network to adjust surface placement of these borders according to features learned from the manual delineation of these boundaries. To achieve this, the following steps were taken:

1. For initializing vertex locations, a midsurface was generated that was closer to the pial surface in sulci and closer to the white surface in gyri, weighting the distance vector by the cortical curvature. Thus, to prevent intersections of subsequent vertex vectors, the midsurface was closer to the surface with the higher curvature.

2. This midsurface was upsampled from 163,842 to 655,362 vertices to increase its resolution.

3. For each vertex, the vector between nearest points on the pial and white surfaces was calculated.

4. To avoid crossing profiles, which can result in mesh self-intersections, the vector components were smoothed across the midsurface with a FWHM of 3 mm (S1 Fig).

5. Profiles were calculated along these vectors from the midsurface, extending the profiles 0.5 mm farther than the minimum distance in the pial or white direction, to ensure the resultant intensity profile captured the full extent of the cortex.

The resulting profiles were less oblique and more likely to be lined up with the cortical columns (S1 Fig).

Extended intensity profiles were then created by sampling voxels at 200 equidistant points between each pair of vertices from the raw and anisotropically smoothed BigBrain volumes, at each available resolution. For an extended profile of approximately 4 mm, this gives a distance of 0.02 mm or 20 μm between points, corresponding to the highest resolution volume available. To account for the rostrocaudal gradient in staining intensity enabling the network to better generalize between profiles, profile intensity values were adjusted by regressing between mean profile intensity and posterior-anterior coordinate in 3D space.

## Training data

Manual segmentations of the 6 cortical layers were created on 51 regions of the cortex, distributed across 13 of the original histological BigBrain sections rescanned at a higher in-plan resolution of 5 μm (Fig 2). These regions were chosen to give a distribution of examples demonstrating a variety of cytoarchitectures, a variety of rostrocaudal locations, in both gyri and sulci, from sections where the cortex was sectioned tangentially.

Layers were segmented according to the following criteria. Layer I, the molecular layer, is relatively cell sparse with few neurons and glia. Layer II, the external granular layer, is a much denser band of small granular cells. Layer III, the external pyramidal layer, is characterized by large pyramidal neurons that become more densely packed toward its lower extent. Layer IV, the internal granular layer (usually referred to simply as the "granular layer"), generally

contains only granular neurons, bounded at its lower extent by pyramidal neurons of layer V. Layer V, the internal pyramidal layer, contains large but relatively sparse pyramidal neurons, whereas layer VI, the multiform layer, has a lower density of pyramidal neurons [1]. Alongside association areas with such typical neocortical laminar structure, samples from the primary visual and motor cortices were specifically included as they exhibit unique laminar characteristics.

Segmentations were verified by expert anatomists: SB, NPG, and KZ. This resolution is sufficient to distinguish individual cell bodies, a prerequisite to analyze their distribution pattern in cortical layers and to delineate the layers. Averaged across all training examples, layer classes contributed to profiles as follows: background/cerebrospinal fluid (CSF): 14.6%, layer I: 7.5%, layer II: 5.6%, layer III: 20.8%, layer IV: 5.5%, layer V: 14.8%, layer VI: 17.8%, white matter: 13.4%. For the cortical layers, these values represent an approximate relative thickness.

Manual segmentations were then coregistered to the full aligned 3D BigBrain space. The manually drawn layers were used to create corresponding pial and white surfaces. These cortical boundaries were extended beyond layer VI and beyond the pial surface between 0.25 mm and 0.75 mm so as to match the variability of cortical extent in the test profile data set. Training profiles were created by sampling raw, smoothed, and manually segmented data, generating thousands of profiles per sample. Each pixel in the labeled data had a class value of 0 to 7, in which pixels superficial to the pial surface were set to 0, followed by layers numbered 1 to 6, and white matter was classed as 7. This 1D profile-based approach greatly expanded the training data set from 51 labeled 2D samples to over 500,000 profiles. Coregistered manually annotated data are available to download at ftp://bigbrain.loris.ca/BigBrainRelease.2015/Layer_Segmentation/Manual_Annotations/.

## Neural network

A 1D convolutional network for image segmentation was created to enable the identification of laminar-specific profile features, which can appear at a range of cortical profile depths [28]. The network was created using stacked identical blocks. Each block contained a batch normalization layer to normalize feature distributions between training batches, a rectify nonlinearity layer used to model neurons, which can have a graded positive activation but no negative response [58], and a convolutional layer [59]. There was a final convolutional layer with filter size 1 and 8 feature maps, 1 for each class. The cost function was median class-frequency-weighted cross-entropy. Class-frequency weighting was added to weigh errors according to the thickness of the layers so that incorrectly classified points in thinner layers were more heavily weighted than errors in incorrectly classified thicker layers [60]. Raw and smoothed profiles were considered as 2 input channels. The network was iteratively trained until the accuracy did not improve for 50 epochs (all training profiles are seen once per epoch). At this point, the previous best model was saved and used for subsequent testing on the full data set. When testing the network, a soft maximum was then applied to detect the most likely layer class for each point. The output was a matrix of 8 (predicted layers) by 200 (sample points) by 655,362 (vertices on a mesh) by 2 (cortical hemispheres).

For each vertex, a measure of confidence was calculated from these predictions. Per-point confidence is the difference between the prediction value for the highest predicted class and the value of the second-highest predicted class. Per class/layer confidence is the mean confidence for all points in that class/layer. The per-vertex summary measure is the mean across all points in the profile. These measures give an indication of the relative confidence for the regional and laminar classifications.

**Hyperparameter optimization and cross-validation.** Here, a set of 50 experiments with random hyperparameters was carried out to explore their impact on training accuracy (there is no consensus method for finding optimum parameters for a neural network). Learning rate, convolutional filter size, number of layers (blocks), weight decay, and batch size were all varied. In summary, the final network was initialized with 6 layers, filter size = 49, learning rate = 0.0005, weight decay = 0.001, in which the learning rate determines the amount weights are updated on each iteration, and weight decay determines the rate at which weights decrease each iteration (which helps prevent overfitting).

For network cross-validation, the manually labeled areas were subdivided into 10 equally sized random subsets or folds. Initially, 2 folds were removed from the data set during training, and network weights were optimized for segmenting samples on one of these folds. This trained network was then used to predict layers on the final previously unseen test fold from which the accuracy was calculated. This process was repeated 10 times to generate an estimate of the network's ability to segment novel cortical regions. The same process was carried out using profiles extracted at all available resolutions.

For generating BigBrain layer segmentations, the network was trained on the full training data set and tested on all intensity profiles.

**Shrinkage estimate.** Histological processing, including fixation and sectioning, causes distortion of the tissue that is nonuniform in the x, y, and z directions. Part of this distortion was corrected in the original reconstruction of the BigBrain [16].

Initial shrinkage of the brain during fixation prior to sectioning was calculated based on the estimated fresh volume of BigBrain, inferred from the original fresh weight, and the volume after histological processing. This gave a volume-based (3D) shrinkage factor of 1.931, which corresponds to an isotropic length-based (1D) shrinkage factor of 1.245.

To estimate the scale of shrinkage in each of the 3 orthogonal directions, the BigBrain volume was linearly coregistered to a volumetric MRI template derived from a group of older subjects (ADNI) [61]. The transformation matrix gave linear scale factors of 1.15, 1.22, and 1.43 in the x, y, and z directions, respectively, with a mean of 1.26. The concordance of these measures of shrinkage suggests that subsequent thickness and length estimates can be adequately corrected for comparison to in vivo measures.

Thus, to approximately compensate for the nonuniform compression of xyz, we transformed the mesh surfaces into MNI space based on the ADNI template. Subsequent thickness analyses were carried out on the transformed meshes. Such compensation for shrinkage is necessary when analyzing cortical thickness gradients on oblique profiles in 3D over the whole brain. Nonlinear corregistration was not applied, as this can lead to localized warping and nonbiological thickness measurements [62].

**Surface reconstruction: Postprocessing 1D profiles.** One-dimensional classified profiles were transformed into mesh layer boundary reconstructions as follows. Transitions between predicted layers were located for each profile and the coordinates of these transitions became vertex locations for the new layer meshes. For the small number of vertices where the network failed (less than 1%), vertex locations were interpolated from the neighboring vertices. Surface indices were smoothed 0.5 mm FWHM across the cortical surface, and 20 iterations of shrinkage-free mesh smoothing were applied to the output surface [63]. This removed nonbiologically high-frequency changes in surface curvature, most commonly due to minor, local misalignment of consecutive 2D coronal sections.

**Cortical thickness, layer thickness.** Cortical thickness was calculated between pial and white cortical surfaces, and laminar thicknesses were calculated between adjacent pairs of cortical surfaces.

**Masking.** Manual masks were created to remove the medial wall and small numbers of vertices located in the large cuts in the anterior temporal cortex (caused by the saw during extraction of the brain from the skull) from subsequent analyses. The allocortex, including parts of the cingulate and entorhinal cortex that do not have 6 layers, were excluded for comparisons of laminar thickness with von Economo measurements.

**Surface-based parcellations.** For comparison, several existing surface-based parcellations of human cortical surfaces (von Economo: [8,64]; Glasser: [33]) were coregistered to the Big-Brain cortical surfaces using an adaptation of the multimodal surface matching approach [65,66]. These parcellations enabled comparison with histological data from von Economo and in vivo cortical thickness measurements from the Human Connectome Project.

**Gradients and processing hierarchies.** Surface labels for the primary visual, auditory, somatosensory, and motor areas were manually delineated on each hemisphere using morphological markers and histological characteristics (S5A Fig). For each system, a larger area containing associated cortical regions was manually delineated and can be viewed mapped to the cortical surface in S5B Fig [10,17,33]. For each vertex within the associated cortical regions, geodesic distance from the primary sensory labels was calculated (S5B Fig) [10,17,33,67].

Code for all of these analyses is made available at https://github.com/kwagstyl

## Supporting information

**S1 Fig. Improving streamline trajectories.** (A) Streamline vectors were smoothed across the cortical surface by varying degrees to assess the impact of smoothing on (1) the number of self-intersections in the pial and white surfaces and (2) the angle between the streamline and the normal vector on the pial and white surfaces. These optimization curves demonstrate that a FWHM of around 2 mm drastically decreases the number of self-intersections and obliqueness of the streamline vectors relative to the pial and white surfaces. (B) Visualizing streamlines against a histological section. Streamlines more closely follow visible cortical columnar trajectories after this improvement (blue) relative to before this streamline vector smoothing process (red). FWHM, full width at half maximum.
(TIF)

**S2 Fig. Layer confidence maps.** Per-vertex confidence is defined as the difference between the prediction value for the highest predicted class and the value of the second-highest predicted class, averaged over the whole profile. This gives an approximation of the reliability of laminar segmentations for the cortex where ground truth manual segmentations have not been carried out. Confidence for suprapial and white matter classes was high throughout the cortex, thus increasing the confidence in overall cortical thickness measures. Layers exhibit relatively consistent confidence maps, with layer IV least confident overall. This pattern matches with visual observations that layer IV is the most difficult to identify. Regional variations in confidence can guide the choice of target regions for future extensions to the training data.
(TIF)

**S3 Fig. Impact of voxel resolution on overall and layer accuracies.** (A) Overall per-point accuracies on withheld test regions calculated using 10-fold validation. Accuracy decreases with decreasing resolution. (B) Mean deviation in depth prediction on test folds between prediction and manually defined layers. Pial/layer I and layer I–II boundaries exhibited the smallest deviations, followed by II/III, with layer III/IV and VI/white boundaries exhibiting larger deviations.
(TIF)

**S4 Fig. Comparison of white matter surfaces generated by the neural network and by placing the white surface at the maximum intensity gradient.** For visual comparison, the surfaces are overlaid on a 2D section, in which manually segmented layers are available. The maximum intensity gradient white surface (red) was identified on lower-resolution data (200 μm). Superimposed on the histology in grayscale is a section of cortex in which 6 layers were manually segmented. The automated blue surface follows the manually delineated gray–white matter boundary, which is determined by the presence of cortical neurons. By contrast, the red (maximum gradient) surface follows a feature that is consistently superficial to the gray–white matter boundary, corresponding to the layer VIa/VIb boundary. This systematic difference highlights the role of using histological expertise when translating across scales and fields to ensure consistent definitions. It also raises an important question on the placement of the white surface in MRI cortical reconstructions, which is placed at the maximum MRI intensity gradient. This gradient is determined predominantly by myelin contrast and therefore influenced by changes in interregional and longitudinal in cortical myelination. Future cortical segmentation algorithms need to be developed with close reference to histological definitions of the gray/white boundary.
(TIF)

**S5 Fig. (A) Manually segmented primary visual (blue), primary auditory (black, partially buried in the lateral sulcus), primary somatosensory (green), and primary motor (yellow) areas, projected onto a heavily smoothed surface.** (B) Manually segmented regions across which cortical and laminar hierarchical thickness gradients were calculated. (C) Geodesic distance across the cortical surface from the primary areas.
(TIF)

**S1 Data. Cortical and laminar thickness values for von Economo areas and corresponding BigBrain areas.**
(CSV)

**S2 Data. Cortical thickness values for MRI data from Human Connectome Project and from corresponding BigBrain areas.**
(CSV)

## Author Contributions

**Conceptualization:** Konrad Wagstyl, Claude Lepage, Paul C. Fletcher, Adriana Romero, Katrin Amunts, Yoshua Bengio, Alan C. Evans.

**Data curation:** Konrad Wagstyl, Claude Lepage, Sebastian Bludau, Katrin Amunts.

**Formal analysis:** Konrad Wagstyl.

**Funding acquisition:** Konrad Wagstyl, Katrin Amunts, Yoshua Bengio, Alan C. Evans.

**Investigation:** Konrad Wagstyl.

**Methodology:** Konrad Wagstyl, Stéphanie Larocque, Guillem Cucurull, Claude Lepage, Joseph Paul Cohen, Lindsay B. Lewis, Hannah Spitzer, Adriana Romero, Karl Zilles, Katrin Amunts, Yoshua Bengio.

**Resources:** Paul C. Fletcher, Yoshua Bengio.

**Software:** Konrad Wagstyl, Stéphanie Larocque, Guillem Cucurull, Claude Lepage, Joseph Paul Cohen, Lindsay B. Lewis, Thomas Funck, Hannah Spitzer, Adriana Romero, Yoshua Bengio.

**Supervision:** Joseph Paul Cohen, Timo Dickscheid, Paul C. Fletcher, Adriana Romero, Katrin Amunts, Yoshua Bengio, Alan C. Evans.

**Validation:** Konrad Wagstyl, Guillem Cucurull, Sebastian Bludau, Nicola Palomero-Gallagher, Lindsay B. Lewis, Karl Zilles.

**Visualization:** Claude Lepage, Hannah Spitzer, Timo Dickscheid, Alan C. Evans.

**Writing – original draft:** Konrad Wagstyl, Alan C. Evans.

**Writing – review & editing:** Konrad Wagstyl, Claude Lepage, Nicola Palomero-Gallagher, Paul C. Fletcher, Adriana Romero, Karl Zilles, Katrin Amunts, Alan C. Evans.

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
