## [Editor Report · Decision Letter 0]

17 Dec 2019

Dear Dr Wagstyl, 

Thank you for submitting your manuscript entitled "BigBrain 3D atlas of cortical layers: cortical and laminar thickness gradients diverge in sensory and motor cortices." for consideration as a Research Article by PLOS Biology.

Your manuscript has now been evaluated by the PLOS Biology editorial staff, as well as by an Academic Editor with relevant expertise, and I am writing to let you know that we would like to send your submission out for external peer review.

However, please note that the editors would like to pursue your manuscript as a Methods and Resources article and not as a Research Article at this point. Please change the article type when re-submitting.

In addition, before we can send your manuscript to reviewers, we need you to complete your submission by providing the metadata that is required for full assessment. To this end, please login to Editorial Manager where you will find the paper in the 'Submissions Needing Revisions' folder on your homepage. Please click 'Revise Submission' from the Action Links and complete all additional questions in the submission questionnaire.

Please re-submit your manuscript within two working days, i.e. by Dec 19 2019 11:59PM.

***Please be aware that, due to the voluntary nature of our reviewers and academic editors, manuscripts may be subject to delays due to their limited availability during the holiday season. Please also note that the journal office will be closed entirely 21st- 29th December inclusive, and 1st January 2020. Thank you for your patience.***

Kind regards,

Gabriel Gasque, Ph.D.,

Senior Editor

PLOS Biology

---

## [Decision Letter · Decision Letter 1]

10 Jan 2020

Dear Dr Wagstyl,

Thank you very much for submitting your manuscript "BigBrain 3D atlas of cortical layers: cortical and laminar thickness gradients diverge in sensory and motor cortices." for consideration as a Methods and Resources at PLOS Biology. Your manuscript has been evaluated by the PLOS Biology editors, by an Academic Editor with relevant expertise, and by three independent reviewers. You will see that reviewer 2, Matthew F Glasser, has signed his comments.

In light of the reviews (below), we are pleased to offer you the opportunity to address the comments from the reviewers in a revised version that we anticipate should not take you very long. We will then assess your revised manuscript and your response to the reviewers' comments and we may consult the reviewers again.

We expect to receive your revised manuscript within 1 month.

**IMPORTANT - SUBMITTING YOUR REVISION**

*NOTE: In your point by point response to to the reviewers, please provide the full context of each review. Do not selectively quote paragraphs or sentences to reply to. The entire set of reviewer comments should be present in full and each specific point should be responded to individually.

*Resubmission Checklist*

*Published Peer Review*

*PLOS Data Policy*

*Blot and Gel Data Policy*

Sincerely,

Gabriel Gasque, Ph.D., 

Senior Editor

PLOS Biology

REVIEWS:

Reviewer #1: This is an interesting manuscript from a well established, expert group with solid credentials in the domains of cortical organization, cytoarchitectonics, and atlas presentation. The manuscript continues and extends related investigations (as, K. Wagstyl et al., Cerebral Cortex, 2018), but can be considered as a significant contribution to this still developing field, and providing specific technical improvements. The latter, as pertain to a comparison of manual vs. automatic segmentation of laminar borders and thickness gradients , and the calculation of laminar thickness over the whole brain. I have only a few comments about issues I think would benefit from author consideration and revision.

1. Primarily, the figures. At first glance, these are attractive, but with closer inspection, I found the figures to be uneven in quality. In Figure 1, the very important issue of laminar borders is not successfully conveyed, largely owing to too low a magnification. The blue of layer II was hard to distinguish from layer I, and the tan of Layer V, with only a few exceptions, was indistinguishable from layer VI. Can the authors 1) modify the color code? and/or 2) re-format the figure. The section outlines could easily be smaller, with more space given to higher magnification insets, where the laminar differentiation (as in Figure 2) can be more fully appreciated. In Figures 3 and 4, the coloration is hard to appreciate (even though closely following that in Wagstyl et al., Cerebral Cortex, 2018). Can the authors consider modifying the contrast or other aspect? 

2. There is a needless lack of clarity regarding "sensory cortices." For most of the manuscript, attention is given to visual and somatosensory, but the authors obviously included auditory cortex in their sensory cortical group (Conclusion, and page 14). 

3. There is some lack of detail; for example, relating to sulcal and gyral components. Area V1 included multiple loci for both. Can this be elaborated on?

4. There is some "over-reaching." This occurs repeatedly in the Discussion: Paragraph 2 "the cortical microcircuit": the Authors have inadequate data (interneurons; EM) for commenting on cortical microcircuitry. Paragraph 2 (page 7) "Likely driver of the laminar and overall thickness gradients" = ? Same, page 8: "cortical microcircuit properties" = ? I suggest that the comparisons with lamination in MRI and Big Brain are all substantive and impotant to state, but some of these other points, in my opinion, are not compelling and thus distracting.

Reviewer #2, Matthew F Glasser: The authors conduct a very interesting analysis of the cortical layers of the big brain dataset, using machine learning to automate the problem and find results that are in agreement with prior datasets and my neuroanatomical expectations. I have a few minor comments that I hope the authors are willing to address. 

1) Was the mid-surface constructed according to the equal volume principle (Bok et al 1929, Van Essen et al 1980 Journal of Comparative Neurology, Waehnert et al 2014 Neuroimage)?

2) To correct for shrinkage, it is said that the big brain was registered to MNI space. Glasser et al 2016 Nature Neuroscience pointed out that MNI space is "drifted" relative to the average of the individual brains prior to registration, leading to an overestimation of 37% brain volume (see supplementary Figure 9). Was this corrected for or avoided somehow or avoid biasing the results presented here?

3) The figures showing overall and laminar cortical thickness might be better displayed using more inflated surfaces so that one can see further inside the sulci. Additionally, the color scales are not very dynamic, making it harder to appreciate the differences across the cortical surface.

4) The authors point out a potential mismatch between MRI-based white matter surfaces and the true anatomical white matter surface based on the mismatch between the big brain's original white matter surface and the improved one using machine learning. While it is true that a maximum intensity gradient is typically used to place the grey/white boundary, the source of this gradient would be different in T1w or T2w MRI versus the cell-body stained histological data here. Thus, it is not a given that MRI surfaces will have the same bias as shown here. Instead, I would recommend the authors explicitly compare group average cortical thickness maps made from high quality T1w and T2w data from the HCP to their gold standard thickness maps to identify any global or local biases. 

Reviewer #3: The authors present a novel application of machine learning to neuroanatomy of the human cerebral cortex. The algorithm used could segment cortical layers based on training data from drawings from human experts. The results focus on assessments of laminar and overall cortical thickness to construct a 3D laminar atlas of the human cerebral cortex. Both layer-specific and overall thickness estimates correlated well with the estimates of von Economo, with only layer 2 measurements showing weak agreement. The authors concluded that somatosensory and visual cortical systems showed opposite thickness gradients to the fronto-motor system.

The constructed atlas of the human cortex at laminar resolution will be useful for others who may use the algorithm to map other findings based on markers to shed light on the organization of the human cortex. Several points throughout the manuscript need clarification. Despite the methodological nature of the paper, the Methods are sketchy and need to be elaborated. It is not even clear what stains the authors used to measure the layers from histological sections. In addition, in view of the exclusion of some areas from the measurements, either by choice or because of methodological difficulties, the caveats should be discussed, and conclusions tempered.

Major comments:

1. p 13, the authors stated: "Manual masks were created to remove the medial wall, the allocortex, including parts of the cingulate and entorhinal cortex which do not have 6 layers, and large cuts in the anterior temporal cortex (caused by the saw during extraction of the brain from the skull) from subsequent analyses."

This is an important point and should be clearly stated in Results as well. It is often difficult to distinguish layer 2 from layer 3, and layer 5 from layer 6 in several association areas. The Results need to be expanded to include commentary on this issue.

2. Related to the above, there is an incorrect statement in the first sentence of the Introduction "The cerebral cortex has six cytoarchitectonic layers..". As classical and modern studies have clearly shown, many cortical areas have fewer than six layers; these include the cortical limbic areas, as shown in the classical studies of von Economo/Koskinas and Sanides, which the authors refer to. More recent studies have relied on the systematic variation of the cortex to derive principles of cortical organization and connections.

3. Page 6, under "Confidence results", "layer IV is often the most difficult to identify", is also related to the point above, because many areas either lack or have a poorly delineated layer IV. The authors unfortunately did not include many of these areas in the analysis, but even so, layer IV varies considerably among association areas that have six layers.

4. In the Abstract: "In contrast, fronto-motor cortices showed the opposite pattern, …", please re-work the sentence to state the direction of "decreases in total and pyramidal layer thickness". 

5. In the Abstract, the authors should state that the BigBrain data are derived from a single postmortem human brain. In view of this fact, the reference to '..a 3D model of the human brain..' should more appropriately be called a 3D atlas. 

6. Defining "geodesic distance" at the first instance of use is necessary for a general audience of readers who may not be familiar with graph theory.

7. One of the limitations of the study that deserves discussion is the sole reliance on laminar and overall cortical thickness. Many architectonic areas have gyral and sulcal parts, and the former are generally thicker than the latter but are not necessarily less dense. The authors have not used density as an additional measure but refer to studies from the literature for discussion of this point. However, the statement that "..neuronal density decreases with increasing cortical thickness [38,39]", (page 7), is not based on detailed histological data of many areas. The use of a limited number of areas, lumping together areas that differ in thickness, or using low-resolution data, may not critically test the generality of a trend or model. This is evident in data on laminar-specific connections from the literature that were advanced as supportive of hierarchical or other models, but held only for a limited set of areas, or held only when some data were excluded from analysis, as was the case in ref. 20. 

Figures:

8. Fig. 1. It is not possible to distinguish layer layer I from II or III with the colors chosen; a brighter color between layers would help.

9. Fig. 3. It would be helpful to identify areas that show the greatest disagreement with the von Economo data. 

10. Figure 4: A table with the quantitative laminar thickness data is needed in Results. A correlation matrix displaying the agreement between the current results and the von Economo findings would be easier to parse than the brain surface heatmaps. Something similar may help with Supplementary Figure 2.

11. Fig. 6A is confusing; does the label under 'motor' include other areas, and if so, which?

Is layer III in motor cortex thicker than layer V, as in D and E? References to "Fronto-motor" (and throughout the manuscript) conjure up the opposite image of the one intended; perhaps it would be better to say motor-frontal.

12. As also mentioned above, the inverse relationship in the motor cortex that the authors describe is partly driven by the areas included in the study. A detailed list of all areas that were included in the analysis is needed. 

13. Supplementary Fig. 4: The figure legend needs work. What is the meaning of the shaded sites/layers in blue/grey?

Minor comments:

14. Page 3, it is not clear what the authors mean by "stereological bias".

15. p 10. The term "diffusive smoothing algorithm" is used, but there is no citation or explanation.

16. p 10. A diagram might help the reader understand how 'vertices' were selected and used.

---

## [Editor Report · Decision Letter 2]

7 Feb 2020

Dear Dr Wagstyl,

Thank you for submitting your revised Methods and Resources article entitled "BigBrain 3D atlas of cortical layers: cortical and laminar thickness gradients diverge in sensory and motor cortices." for publication in PLOS Biology. I have now discussed your revision with the staff editors and with the Academic Editor as well. I'm delighted to let you know that we're editorially satisfied with your manuscript. However before we can formally accept your paper and consider it "in press", we also need to ensure that your article conforms to our guidelines. A member of our team will be in touch shortly with a set of requests. As we can't proceed until these requirements are met, your swift response will help prevent delays to publication. Please also make sure to address the data and other policy-related requests noted at the end of this email.

*Copyediting*

*Published Peer Review History*

*Early Version*

*Submitting Your Revision*

Sincerely,

Gabriel Gasque, Ph.D., 

Senior Editor

PLOS Biology

DATA POLICY:

Please ensure that the figure legends in your manuscript include information on where the underlying data can be found, and ensure your supplemental data file/s has a legend.

---

## [Editor Report · Decision Letter 3]

18 Mar 2020

Dear Dr Wagstyl,

On behalf of my colleagues and the Academic Editor, Henry Kennedy, I am pleased to inform you that we will be delighted to publish your Methods and Resources in PLOS Biology. 

Early Version

PRESS 

Kind regards,

Alice Musson

Publication Assistant, 

PLOS Biology

on behalf of

Gabriel Gasque,

Senior Editor

PLOS Biology